

# Technical note: Adaptably diagnosing O₃-NOₓ-VOC sensitivity evolution with routine pollution and meteorological data

Minjuan Huang [1,2] *, Tengchao Liao [1,2]

[1] School of Atmospheric Sciences, Sun Yat-sen University, and Southern Marine Science and Engineering Guangdong
Laboratory (Zhuhai), Zhuhai 519082, PR China
[2] Guangdong Provincial Observation and Research Station for Climate Environment and Air Quality Change in the Pearl River
Estuary, Key Laboratory of Tropical Atmosphere-Ocean System (Sun Yat-sen University), Ministry of Education, Zhuhai,
519082, PR China

*Correspondence to*: Minjuan Huang (hminjuan@mail.sysu.edu.cn)

**Abstract.** Elucidating the evolving O₃-NOₓ-VOC sensitivity in response to varying precursor emission trends is critical for
mitigating the elevating ozone. Due to the complexities and resource constraints inherent in conventional methods, we
developed an adaptable methodology addressing this issue through empirical parametric regression of routine data
(O₃/NOₓ/NO₂). The *log-Bragg3* model (Equation 3) performed best in globally characterizing the daytime ozone production
(DPO₃)-NOₓ (or NO₂) relation, including regions with severe PM₂.₅ contamination where ozone formation is additionally
influenced by aerosol-inhibited photochemical regime. Over 95% of these fits achieved statistical significance (p<0.1). This
model provides parametric interpretations of ozone formation intensity (*d*), the associated chemical processes (*b*), and the O₃-
NOₓ-VOC sensitivity partition threshold (*e*). More vigorous photochemical reactions are implicated in the studied Chinese
regions by higher values of parameters *b* (0.87-2.42) and *d* (34.72-54.78) relative to EU/US (*b*=0.26-0.57, *d*=9.97-31.45).
Divergent temporal trends in parameter *b* further indicate fundamentally distinct evolutionary pathways in regional ozone
chemistry between China and EU/US. Specific to MDA8-daytime hours, the Chinese city agglomerations were all diagnosed
as being in the VOC-limited regime in both 2014 and 2019 on the regional scale, exhibiting significantly higher spatial
predominance than the previous satellite-derived HCHO/NO₂ ratio inferences. The DPO₃-NO₂ pseudo-diagnosis constituted
major uncertainty in spatiotemporal diagnosis, whereas the DPO₃-NOₓ curve showed superior reliability. This methodology
helps provide critical insights for formulating spatially differentiated precursor control policies.



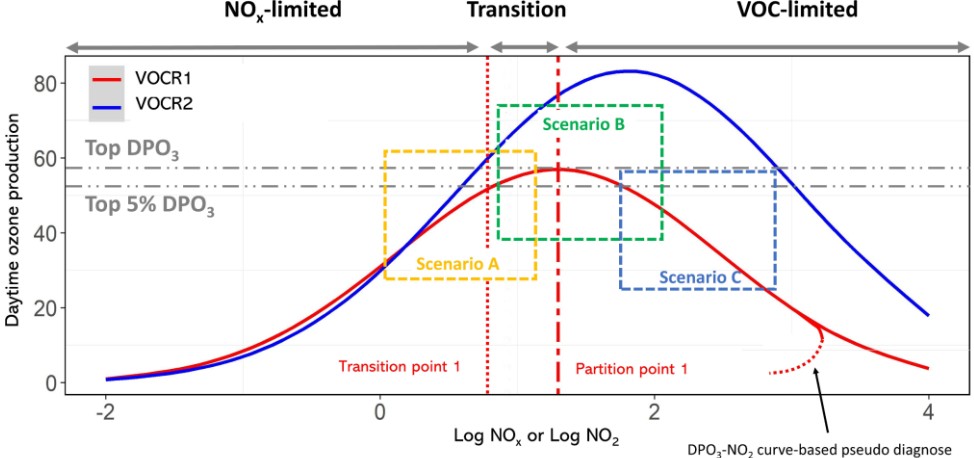

**Graphical Abstract**



# 1 Introduction

Tropospheric surface ozone ($O_3$) pollution harms human health and ecosystem (Collaborators, 2020; Ban et al., 2022;
Agathokleous et al., 2023; Agathokleous et al., 2020; Feng et al., 2022). It is rapidly produced from the sunlight-driving
oxidation of carbon monoxide (CO), methane ($CH_4$), and volatile organic compounds (VOCs) with the presence of nitrogen
oxides ($NO_x=NO+NO_2$) (Atkinson, 2000; Wang et al., 2017; Li et al., 2018). Owing to shifting precursor emissions (Vazquez
Santiago et al., 2024; Zhang et al., 2023), declining aerosol contamination (Li et al., 2019; Ivatt et al., 2022) and the warming
environment (Schnell and Prather, 2017; Xiao et al., 2022b; Guo et al., 2023), the ozone level has increased in most urban
areas worldwide (Wang et al., 2024; Vazquez Santiago et al., 2024). The $O_3$-$NO_x$-VOC sensitivity has likely evolved in
response to the divergent trends in precursor emissions, and elucidating its long-term evolution is critical for effective ozone
mitigation.

The current diagnosis methods, such as the Empirical Kinetic Modeling Approach (EKMA) isopleth plot and chemical
indicators (e.g., $H_2O_2/HNO_3$, $H_2O_2/NO_z$, etc.), heavily rely on observation-based or numerical models, constrained by limited
field data and computational demands. They are typically applied in case studies (Sillman and He, 2002; Sillman and West,
2009; Xue et al., 2014; Ou et al., 2016; Li et al., 2018). Although the satellite-derived $HCHO/NO_2$ ratio (FNR)-based method
enables the regional scale long-term $O_3$-$NO_x$-VOC sensitivity diagnosis (Jin and Holloway, 2015; Ren et al., 2022; Wang et
al., 2021; Zhang et al., 2024), its fixed daily sampling time restricts insights into other hours, and the sensitivity always varies
over time (Sillman and West, 2009). These constraints highlight the necessity for more flexible and adaptable approaches.

For a specific VOC reactivity (VOCR), the daytime ozone production ($DPO_3$) exhibits a characteristic skewed curve when
plotted against $NO_x$ or $NO_2$, which is transformed from the EKMA plot (Pusede and Cohen, 2012; Romer et al., 2018;
Nussbaumer and Cohen, 2020; Guo et al., 2023; Yang et al., 2021). As shown in the Graphical Abstract, ozone production
rises with $NO_x$ is insensitive to VOCR. Reducing $NO_x$ is more effective than controlling VOCs to mitigate ozone. As $NO_x$
increases. As $NO_x$ rises, it reaches its maximum and became limited by both $NO_x$ and VOCR. This indicates that controlling
either or both precursors can effectively reduce ozone acceleration. With further $NO_x$ increase, it grows with VOCR but
declines with $NO_x$. Here, VOC control becomes the key for ozone mitigation, while $NO_x$ reduction potentially leads to increase
in ozone pollution.

Non-parametric smoothing of this curve using routine monitoring data has proven effective for OFR diagnosis in the
Guangdong-Hong Kong-Macao Greater Bay Area, South China (Huang et al., 2025). However, two critical limitations persist:
(1) a fixed smoothing configuration fails to exhibit robustness in fitting performance across spatiotemporal scales; (2) it fails
to identify the $NO_x$-limited/transition boundary (red dotted line in Graphical Abstract), that requires parametric modelling. The
$DPO_3$-$NO_x$ (or $NO_2$) curve shows environmental stability (Guo et al., 2023), enabling the parametric characterization. Unlike
the $DPO_3$-$NO_x$ curve, the $DPO_3$-$NO_2$ curve may theoretically exhibit a bend in cases with extremely low $DPO_3/NO_2$ ratio



(Graphical Abstract) (Romer et al., 2018; Guo et al., 2023); in such cases, the ozone production decreases with $NO_2$, leading
to a pseudo-diagnosis of the $NO_x$-limited regime under a realistic $NO_x$-saturated condition., which has been observed in Hong
Kong (Huang et al., 2025).

Based on the above premises, the present study aims to: (1) verify the universality of $DPO_3$-$NO_x$ (or $NO_2$) relation using
routine monitoring networks (Section 3.1); (2) identify a globally capable empirical model for $DPO_3$-$NO_x$ (or $NO_2$)
characterization (Section 3.2); (3) compare the reliability between the $DPO_3$-$NO_2$ and $DPO_3$-$NO_x$ curves in diagnosing $O_3$-
$NO_x$-VOC sensitivity (Section 3.3); by utilizing the identified empirical model, (4) investigate the evolution of $O_3$-$NO_x$-VOC
sensitivity for four Chinese city agglomerations (the Beijing−Tianjin−Hebei and surrounding (BTH) region, the Fenwei Plain
(FWP), the Yangtze River Delta (YRD) region, and the Greater Bay Area (GBA)), the European Union (EU) and the United
States (US), as well as discuss the model parametric implications and the primary uncertainty source for spatiotemporal
diagnosis (Sections 3.4-3.6). We focused on the $DPO_3$−$NO_x$ (or $NO_2$) relation and $O_3$-$NO_x$-VOC sensitivity within the MDA8-
daytime hours, as MDA8 is one of the key air quality standard metrics around the world and usually employed to examine the
ozone exposure attributable human health adverse effects.

## 2 Methodology

The $DPO_3$-$NO_x$ (or $NO_2$) relation was regressed with the five-percentile-binned $NO_x$ (or $NO_2$) concentrations (or
logarithms) and their corresponding average $DPO_3$ levels. The $DPO_3$ was defined as the difference between the MDA8-daytime
(7:00-19:00 Local Time (LT)) hourly ozone concentration and the ozone concentration at 6:00 LT. The non-parametric
regression, smoothing a numerical series in local neighbourhood, was firstly utilized to reveal the intrinsic $DPO_3$-$NO_x$ (or $NO_2$)
relation. The parametric model validity was confirmed when its curve and partition point aligned well with the non-parametric
results, demonstrating accurate ozone formation regime diagnosis.

### 2.1 Parametric models

A total of seven parametric models were individually applied to characterize the $DPO_3$-$NO_x$ (or $NO_2$) relation, including the
five-parameter *Beta* (Equation 1) and *logarithmic Beta* (*log-Beta*) (Equation 2) functions, the three- and four-parameter
*logarithmic Bragg* (*log-Bragg3 and log-Bragg4*) functions (Equations 3-4), the three- and four-parameter *logarithmic Lorentz*
(*log-Lorentz3 and log-Lorentz4*) functions (Equations 5-6), and the *logarithmic quadratic polynomial* (*log-Poly2*) function
(Equation 7), where *Y* represents $DPO_3$ levels and *X* denotes $NO_x$ (or $NO_2$) concentrations.

$$Y = d \times \left\{ \left( \frac{X - x_b}{x_o - x_b} \right) \times \left( \frac{x_c - X}{x_c - x_o} \right)^{\frac{x_c - x_o}{x_o - x_b}} \right\}^b \tag{1}$$





$$Y = d \times \left\{ \left( \frac{log(X)-log(x)_b}{log(x)_o-log(x)_b} \right) \times \left( \frac{log(x)_c-log(X)}{log(x)_c-log(x)_o} \right)^{\frac{log(x)_c-log(x)_o}{log(x)_o-log(x)_b}} \right\}^b \qquad (2)$$

Specifically, in *Beta* and log-Beta models (Equations 1-2), parameter $b$ determines curve shape; $d$ represents the maximum fitted $DPO_3$; $x_b$ (or $log(x)_b$) denotes the $NO_x$ (or $NO_2$) concentration (or its logarithm) at peak $DPO_3$, serving as the partition point; $x_o$ (or $log(x)_o$) and $x_c$ (or $log(x)_c$) define the regression's lower and upper x-axis boundaries, respectively.


$$Y = d \times exp[-b \times (log(X) - e)^2] \qquad (3)$$

$$Y = c + (d - c) \times exp[-b \times (log(X) - e)^2] \qquad (4)$$

$$Y = \frac{d}{1+b \times (log(X)-e)} \qquad (5)$$

$$Y = c + \frac{d-c}{1+b \times (log(X)-e)} \qquad (6)$$

In *log-Bragg3, log-Bragg4, log-Lorentz3 and log-Lorentz4* models (Equations 3-6), $b$ also relates to the curve width, determining the location of transition point; $d$ also represents the maximum $DPO_3$ level; $e$ is the maximum $DPO_3$ corresponding $NO_x$ (or $NO_2$) level (the partition point); $c$ is the non-zero lower asymptote.

$$Y = b_0 + b_1 \times log(X) + b_2 \times log(X)^2 \qquad (7)$$

In *log-Poly2* model (Equation 7), the parameters $b_0$, $b_1$ and $b_2$ do not have clear theoretical meanings.

**2.2 Data sources and pre-processing**

The observed hourly concentrations of $O_3$, $NO_2$ and $NO_x$ (2014-2019) for the regions of BTH, FWP, YRD, PRD and Hong Kong in China, as well as the US and EU were employed to investigate the parameterization of the $DPO_3$-$NO_x$ (or $NO_2$) relation. Data were sourced from the Ministry of Ecology and Environment of the People's Republic of China
(https://data.epmap.org), the Environmental Protection Department of Hong Kong (https://cd.epic.epd.gov.hk), the United States Environmental Protection Agency (https://aqs.epa.gov/aqsweb/airdata/download_files.html#Raw), and the European Environment Agency (discomap.eea.europa.eu/map/fme/AirQualityExport.htm), respectively. Only stations with more than 75% completeness of recordings per year or during the entire study were included in this study. For Macao, the data were obtained from the high-resolution Chinese air quality reanalysis dataset (CAQRA, 15 km)
(https://www.scidb.cn/cstr/31253.11.sciencedb.00053). The pollutant concentration was expressed as mixing ratio (ppb).

In order to avoid the cleanup effects on the $DPO_3$-$NO_x$ (or $NO_2$) relation caused by transport, ventilation and deposition, only the hours with calm wind ($\leq 3.3$ m/s) and "no precipitation" were incorporated. The hourly meteorological datasets used for pollution data screening included the National Oceanic and Atmospheric Administration (NOAA) Integrated Surface



Database (ISD) (ftp://ftp.ncdc.noaa.gov/pub/data/noaa/isd-lite), the CAQRA dataset, and the European Centre of Medium-

range    Weather    Forecasts    Reanalysis    v.5    (ERA5)    global    reanalysis    datasets    (31    km)
(https://www.ecmwf.int/en/forecasts/dataset/ecmwf-reanalysis-v5).

For the EU and US, the "no precipitation" scenario was defined as the hours of 50% cloud cover with the records ≤ 4 in ISD,
rather than the zero-precipitation hours, due to the limited rainfall recordings in the ISD compared to cloud cover, especially
in the US. The wind speed and cloud cover data for a specific pollutant station were derived from the recordings at its nearest

surrounding ISD station(s) located within 15 km and 50 km, respectively. Pollution monitoring stations without matching
meteorological information were excluded.

For the studied regions in China, the ISD recordings of both precipitation and cloud cover were sparse, and the number of
meteorological stations were insufficient for regional representation (BTH: 11 stations, FWP: 9 stations, YRD: 29 stations,
GBA: 5 stations). In order to retain as much data as possible, the windspeed and precipitation data were extracted from CAQRA

and ERA5 datasets, respectively. The CAQRA gridded screening wind speed was adjusted from 3.3 m/s to 6 m/s, based on the
linear regression relation between the ISD observation stations and their corresponding CAQRA gridded records. In these
regions, the "no precipitation" scenario was exactly defined as the zero-precipitation (< 2mm) hours, based on the ERA5
gridded rainfall records. The data processes and visualization in this study were conducted with R and Python.

A total of 493 stations (14 in EU, 114 in US, 70 in BTH, 55 in FWP, 169 in YRD, 71 in GBA (57 in PRD, 14 in Hong

Kong)), and 1 CAQRA grid (for Macao) were incorporated to compare the capabilities of different parametric models (Figure
S1). A total of 1306, 813, 1814 and 305 CAQRA grids were respectively employed to investigate the spatiotemporal variations
of $O_3$-$NO_x$-VOC sensitivity from 2014 to 2019 for the regions of BTH, FWP, YRD and CBA in China.

## 3 Results and discussion

### 3.1. The DPO$_3$−NO$_x$ (or NO$_2$) relation is empirically validated worldwide.

Based on the non-parametric approach, almost all pollution monitoring stations (including the Macao grid) during 2014-
2019 were able to be characterized as part of a theoretical DPO$_3$-NO$_x$ (or NO$_2$) diagram corresponding to the Scenario A, B or
C (Graphical Abstract), except three stations in US. This indicates that such a regular diagram is globally prevalent, even in
regions with severe PM$_{2.5}$ contamination, where the ozone formation is additionally influenced by the aerosol-inhibited
photochemical regime, such as BTH, FWP and YRD in China (Ivatt et al., 2022; Geng et al., 2021; Kong et al., 2021; Xiao et

al., 2022a). Based on the seven studied models (Equations 1-7), the parametric fitting curves were generally consistent with
their corresponding non-parametric smoothened curves on the regional scale (Figures S2-S10). To further investigate their
global fitting capabilities, these models were individually applied to regress the DPO$_3$-NO$_x$ (or NO$_2$) relation for all the studied
stations and the Macao grid.





## 3.2 Which is the most capable parametric model?

All 494 parametric fits converged using the models of *log-Bragg3, log-Bragg4, log-Lorentz3* and *log-Lorentz4* (Equations 3-7). However, based on the *Beta* and *log-Beta* models (Equations 1-2), a total of 59 and 69 out of the 494 DPO$_3$-NO$_2$ fits failed to converge, respectively, for all studied stations; while 17 and 21 out of the 142 DPO$_3$-NO$_x$ fits did not converge in Hong Kong, EU and US. Notably, not all the convergent fits were able to characterize a regular diagram to effectively partition O$_3$-NO$_x$-VOC sensitivity. The amounts of the convergent and effective fits varied across the studied models: *log-poly2* (142

out of 142 DPO$_3$-NO$_x$ fits, 494 out of 494 DPO$_3$-NO$_2$ fits) > *log-Bragg3* (141/142, 490/494) > *log-Lorentz3* (141/142, 489/494) > *log-Bragg4* (140/142, 488/494) > *log-Lorentz4* (122/142, 419/494) > *log-Beta* (121/142, 425/494) > *Beta* (114/142, 405/494). Although all the *logarithmic Poly2* fits (Equation 7) were convergent and effective, quite certain portion of them did not achieve the statistical significance ($p > 0.1$) (Figures S11-S12 (g)). Amongst all models, the *log-Bragg3* and *log-Lorentz3* models performed the best, with over 95% of fits achieving the statistical significance ($p < 0.1$) (Figures S11-S12 (c,

e)). Detailed comparisons of the models' fitting performances are provided in Supporting Information (Text S1).

   By excluding all non-convergent and irregularly convergent fits, as well as fits corresponding to Scenario A or C as shown in Graphical Abstract (52 DPO$_3$−NO$_2$ fits and 18 DPO$_3$−NO$_x$ fits), a total of 442 DPO$_3$−NO$_2$ fits and 124 DPO$_3$−NO$_x$ fits were incorporated for comparing the partition points identified using the non-parametric and parametric approaches (Figure 1). For the DPO$_3$−NO$_2$ relation (Figure 1 (a-g)), the *Beta*, *log-Beta*, *log-Bragg3*, *log-Bragg4*, and *log-Lorentz3* models show strong

alignment ($r^2 \geqslant 94\%$, slopes: 0.84-0.89) (Figure 1 (a-e)). However, only the *log-Bragg3* and *log-Lorentz3* models (Figure 1 (c, e)) identified the partition points for all fits under Scenario B (Graphical Abstract). Similarly, for the DPO$_3$−NO$_x$ relation, these two models (Figure 1 (j, l)) also exhibit the best performances.

   Despite comparable performance in terms of amounts of convergent and effective fits, fitting statistical significance, and ability to identify partition point between the *log-Bragg3* and *log-Lorentz3* models, the *log-Bragg3* model is preferred due to

the generally inferior statistical properties exhibited by *Lorentz* models (Ratkowsky, 1990).





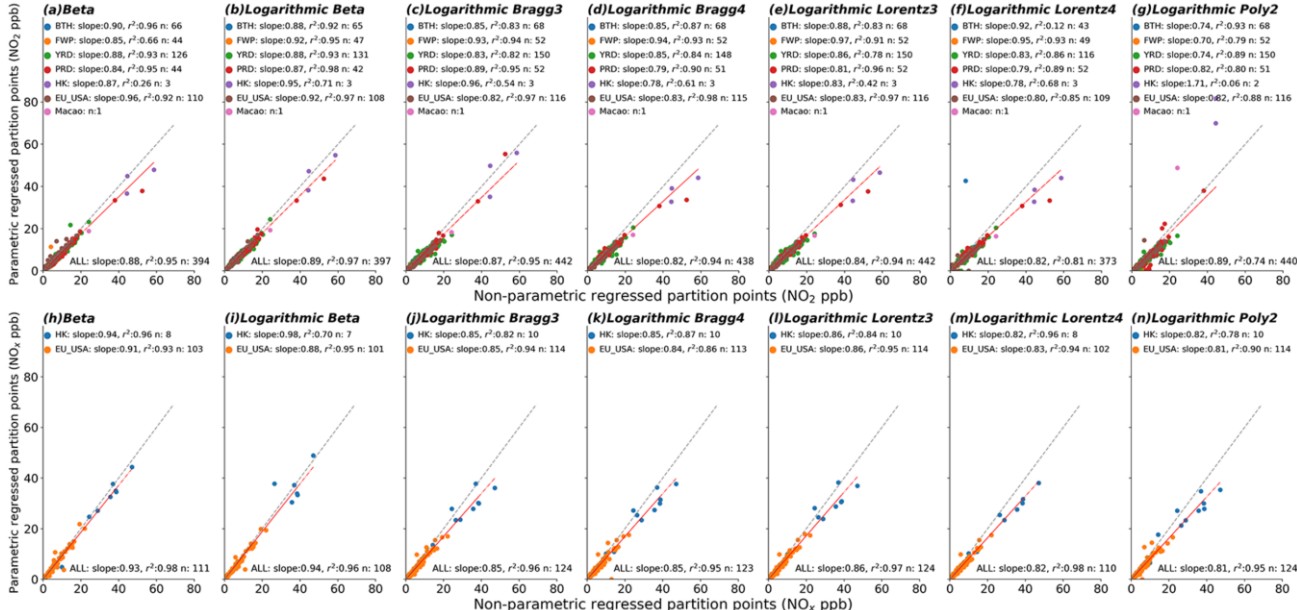

**Figure 1: Comparisons of the partition points recognized between the parametric (y-axis) and non-parametric (x-axis) approaches based on the DPO$_3$−NO$_2$ (a-g) and DPO$_3$−NO$_x$ (h-n) relations, respectively.**

### 3.3 Comparison of reliabilities between the DPO$_3$-NO$_2$ and DPO$_3$-NO$_x$ curves

The proportion of data points with NO$_x$ (or NO$_2$) concentrations on the right of the partition points (referred to as VOC-limited % below) correlated well between the *log-Bragg3* model and the non-parametric method for both DPO$_3$-NO$_2$ and DPO$_3$-NO$_x$ curves (Figure 2 (a, b)). Notably, the agreement on DPO$_3$-NO$_2$ partition points was poor in Hong Kong (Figure 1 (c)), while it was improved for DPO$_3$-NO$_x$ relation (Figure 1 (j)). Most DPO$_3$-NO$_x$ fits (n=10) in Hong Kong aligned with Scenario B, featuring a clear partition point, whereas most DPO$_3$-NO$_2$ fits (n=8) corresponded to Scenario A (Graphical

Abstract). This confirms differing performances of these two curves in diagnosing O$_3$-NO$_x$-VOC sensitivity. Fortunately, the VOC-limited % derived from both DPO$_3$-NO$_2$ and DPO$_3$-NO$_x$ curves agreed well across EU and US; and only some stations in Hong Kong showed the conflicting results: a VOC-limited regime (VOC-limited % > 50%) based on DPO$_3$-NO$_x$ fit v.s. a NO$_x$-limited/transition regime (VOC-limited % = 0) based on DPO$_3$-NO$_2$ fit (Figure 2 (c, d)).

The DPO$_3$/NO$_2$ ratios at these stations in Hong Kong ranged from 0.1 to 0.6, much lower than other stations/grid (BTH: 1.1-

4.0, FWP: 1.3-3.4, YRD: 1.4-4.5, PRD: 1.3-4.5, Macao: 2.1, EU_US: 0.3-16.5, other stations in Hong Kong: 0.8-6.5). A low DPO$_3$/NO$_2$ ratio typically indicates a condition that the reaction of OH with NO$_2$ dominates the fate of HO$_x$, slowing the oxidation of organic precursor, and gradually terminating the ozone production (Pusede et al., 2015; Romer et al., 2018). The ozone production decreases with NO$_2$ under this condition, leading to a pseudo diagnostic result indicative of a NO$_x$-limited regime under a realistic NO$_x$-saturated condition (Guo et al., 2023; Romer et al., 2018; Pusede et al., 2015). Hence, the DPO$_3$-





NO$_x$ curve is considered more reliable for diagnosing O$_3$-NO$_x$-VOC sensitivity at any NO$_x$ level, and it is recommended to check the DPO$_3$/NO$_2$ ratio before employing the DPO$_3$-NO$_2$ curve.

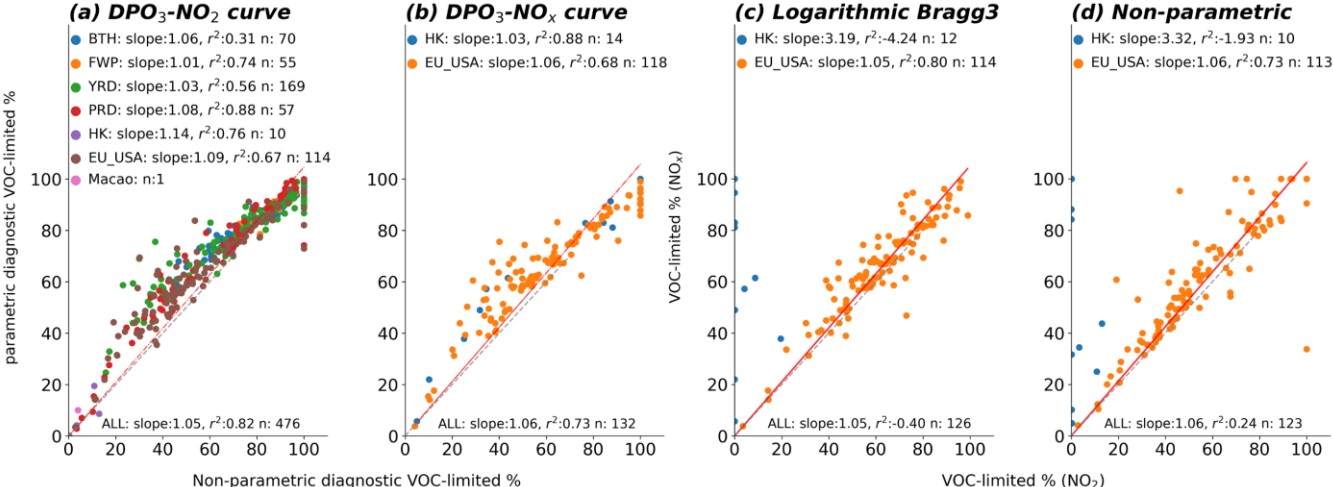

**Figure 2:** Comparisons of the VOC-limited % diagnosed between the *log-Bragg3* model (y-axis) and non-parametric approach (x-
axis) based on the DPO$_3$-NO$_2$ (a) and DPO$_3$-NO$_x$ (b) relations, respectively, and between the DPO$_3$-NO$_2$ (y-axis) and DPO$_3$-NO$_x$ (x-
axis) fits with the *log-Bragg3* model (c) and non-parametric approach (d), respectively.

**3.4 The O$_3$-NO$_x$-VOC sensitivity evolution on the regional scale**

The evolution of O$_3$-NO$_x$-VOC sensitivity (or OFR) was diagnosed based on the variations in the DPO$_3$-NO$_2$ curves between 2014 and 2019 using the *logarithmic Bragg 3* model (Figures 3 and S13). The O$_3$-NO$_x$-VOC sensitivity (or OFR) for a specific
region/station/grid was diagnosed as a NO$_x$-limited regime, transition regime and VOC-limited regime, if the average NO$_2$ concentration was lower than the transition point, between the transition and partition points, and higher than the partition point, respectively. The parametric partition point corresponded to the parameter *e*. The parametric transition point (blue dotted line in Figures 3 and S13) was defined as the NO$_2$ level at the maximum slope midpoint of fitting curve, corresponding to DPO$_3$ level in the top 4.9% of *log-Bragg3* model predictions. The non-parametric transition point was determined as the NO$_2$
level corresponding to the top 4.9% smoothing DPO$_3$ level (red dotted line in Figures 3 and S13).

The observed regional DPO$_3$/NO$_2$ ratios were 1.78, 1.63, 2.18, 1.77 and 3.86 (EU: 0.88, US: 4.03) for BTH, FWP, YRD, GBA and EU/US in 2014, and increased to 2.49, 2.45, 2.71, 2.33 and 4.77 (EU: 1.05, US: 4.98) in 2019, respectively. These generally median ratio levels suggest that the DPO$_3$-NO$_2$ curve is likely to effectively diagnose the OFR evolution on the regional/national scale. All studied regions were consistently under the VOC-limited regime in both years. However, it is
noteworthy that the annual VOC-limited % in BTH, FWP, GBA, EU and US experienced varying declines from 2014 to 2019 (Table S1). This trend of OFR evolving to a less NO$_x$-saturated condition in recent years has also been reported for other city clusters worldwide (Vazquez Santiago et al., 2024). Conversely, the YRD region showed a slight shift towards more NO$_x$-



saturated from 2014 (annual VOC-limited %: 72.10 %) to 2019 (84.63 %), despite a decline in its average $NO_2$ level; this could be attributed to a more pronounced decline in partition point (Figure 3 (c, h), Table S1), resulting in more scenarios

falling to the right of the partition point in 2019. The CAQRA gridded data-based regional diagnosis showed the similar results for the four Chinese city agglomerations (Figure S13, Table S2).

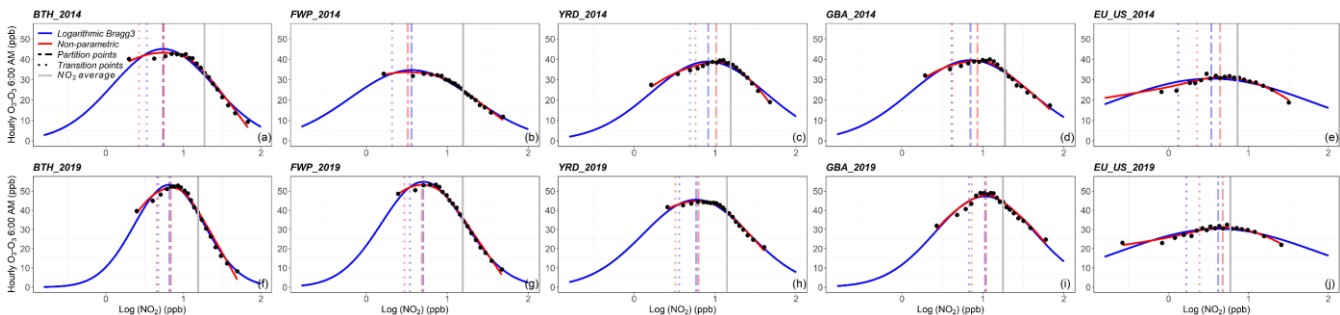

**Figure 3: Variations of the DPO₃-NO₂ curves from 2014 (a-e) to 2019 (f-j) on the regional scale based on the observation data.** The
diagnostic results for both EU and US agreed well between the DPO₃-NO₂ and DPO₃-NOₓ fits, so their DPO₃-NOₓ curves were not additionally provided.

### 3.5 Implications of the *log-Bragg 3* model's parameters (*b, d*)

Based on the observation data, the parameter *d,* that represents the maximum DPO₃, exhibited higher for the studied regions in China compared with the EU/ US; additionally, the parameter *b* in China was consistently higher in 2019 compared to 2014

(Table S1). This indicates that the regional ozone production in China was more intense and the ozone pollution deteriorated from 2014 to 2019, consistent with the findings in previous studies (Lu et al., 2018; Lu et al., 2020). In theory, at low $NO_x$ level, increasing NO enhances the reactions of $RO_2$+NO and $HO_2$+NO, thereby promoting ozone production (Equation 8). As $NO_x$ increases, the $NO_x$ consumption (Equations 9-10) gradually rises, leading to a reduction in ozone production efficiency. At high $NO_x$ level, the production of $HNO_3$ (Equation 10), accelerates significantly and constitutes the predominant $NO_x$

consumption pathway and HO fate, thereby gradually suppressing ozone production (Romer et al., 2018; Pusede et al., 2015; Farmer et al., 2011). The higher value of parameter *b* observed in China, compared with the EU/US (Table S1), characterizes a steeper curve, and indicates a faster change in ozone production efficiency for a given increment of $NO_x$. In addition, the distinct trends in temporal variation of parameter *b* between the regions in China (increasing from 2014 to 2019) and EU/ US (decreasing from 2014 to 2019) (Table S1) imply that they were experiencing different variations in ozone formation

chemistry.

$$PO_3 = (1 - \alpha)k_{NO+RO_2}[NO][RO_2] + k_{NO+HO_2}[NO][HO_2] \qquad \text{(Equation 8)}$$

$$P\sum RONO_2 = \alpha k_{NO+RO_2}[NO][RO_2] \qquad \text{(Equation 9)}$$

$$PHNO_3 = k_{OH+NO_2}[OH][NO_2] \qquad \text{(Equation 10)}$$





### 3.6 Spatiotemporal variation diagnosis of $O_3$-$NO_x$-VOC sensitivity and associated uncertainty

In BTH and FWP, almost all stations and grids were diagnosed as being under the VOC-limited regime in both 2014 and 2019; in most areas of YRD and GBA, the ozone formation were also sensitive to VOC, except those tiny coastal zones (Figure S14). The VOC-limited areas diagnosed in our study were much dominant than those recognized using the satellite-derived $HCHO/NO_2$ ratio (FNR)-based method (Jin and Holloway, 2015; Wang et al., 2021; Ren et al., 2022; Zhang et al., 2024). This discrepancy can be attributed to the differences in studied periods and daily hours, as well as the bias in the satellite-based method. On one hand, our study focused on the entire year and MDA8-daytime hours, while the satellite-based studies covered the ozone-season midday (13:00-14:00 LT), when higher temperatures make ozone formation more $NO_x$-sensitive (Pusede et al., 2014; Pusede et al., 2015; Yang et al., 2021; Huang et al., 2025).This can be further supported by the ozone-season midday (April-September, 13:00-14:00 LT) specific diagnostic results as shown in Figure S15, in which the VOC-limited areas were notably reduced for BTH, YRD and GBA. The satellite-derived column FNR is likely to represent the average condition within the column height, rather than the near-surface condition. The concentrations of ozone and its precursors,  as well as the FNRs, vary with heights (Liao et al., 2021; Mo et al., 2022; Li et al., 2024; Li et al., 2020), and the near-surface environment had been reported to be more VOC-limited (Li et al., 2024). Furthermore, the satellite-derived HCHO cannot differentiate the primary emitted and secondary produced HCHO, whereas the latter is more indicative of the VOC reactions (Liu et al., 2021). Therefore, the satellite-based method is likely to overestimate the near-surface condition, leading to some VOC-limited conditions being misdiagnosed as in the $NO_x$-limited or transition regime.

Beyond Hong Kong, the $DPO_3$-$NO_2$ pseudo-diagnosis was also explicitly identified at some stations in BTH and YRD, highlighted with red circles in Figure 4. The VOC-limited % were recognized near zero (e.g., < 0.1 %) at those stations with the $DPO_3/NO_2$ ratios from 0.31 to 0.84 in 2014; and increased to above 65% with the notably higher ratios (1.91-3.17) in 2019. As discussed before, the $DPO_3$-$NO_2$ curve is effective for OFR diagnosis only if the $DPO_3/NO_2$ ratio is not too low. However, it is difficult to figure out the threshold for the "lowest" ratio, as it varies with studied periods and sites. For instance, the $DPO_3$-$NO_2$ curve could still be reasonably applied at some stations in YRD in 2014, where the $DPO_3/NO_2$ ratios and the diagnostic VOC-limited % ranged from 0.36 to 0.78 and from 80% to 100%, respectively. Fortunately, such pseudo-diagnosis was not found for either the stations or grids in both FWP and PRD (Figures 4 (d-f, j-l) and S14 (c-d, g-h)). The spatiotemporal diagnostic results in Hong Kong were based on the $DPO_3$-$NO_x$ curve.

The VOC-limited % for most areas in FWP experienced varying declines from 2014 to 2019 (Figure 4 (d-f)). For GBA, the areas with the highest VOC-limited % in 2014 were located along the cities from the eastern Pearl River Estuary (PRE) to the northwest of the Bay Area, where the VOC-limited % declined the most by 2019; conversely, the VOC-limited % increased



in the surrounding areas that were originally more $NO_x$-limited in 2014, especially in the northern and eastern areas (Figure 5
(j-l)).

**Figure 4: Spatiotemporal variations of the VOC-limited % from 2014 to 2019 in four city agglomerations of China, based on the *logarithmic Bragg 3* model fitting DPO₃-NO₂ curves.**



## 4. Conclusion

The identified empirical model, *log-Bragg3* model (Equation 3), adaptably resolves the $O_3$-$NO_x$-VOC sensitivity continuum, and provide parametric insights into ozone formation intensity (*d*), associated chemical processes (*b*), and the $O_3$-$NO_x$-VOC sensitivity partition threshold (*e*). More vigorous photochemical reactions are implicated in the studied Chinese regions by higher values of parameters *b* and *d* relative to EU/US. Divergent temporal trends in parameter *b* further indicate fundamentally distinct evolutionary pathways in regional ozone chemistry between China and western regions. Compared with previous

satellite-derived FNR-based studies, our diagnostic results demonstrated a higher prevalence of VOC-limited regimes. This is because our findings reflect the near-surface conditions specific to MDA8 daytime hours throughout an entire year, rather than the average conditions within the satellite column at midday during the ozone season. This parametric methodology overcomes the complexities and resource constraints inherent in conventional methods, and is expected as a unified tool to facilitate global ozone mitigation under evolving precursor emission patterns and climate change. However, the $DPO_3$-$NO_2$ curve-based

pseudo-diagnosis can introduce significant uncertainty in cases of extremely low $DPO_3/NO_2$ ratio. In contrast, the $DPO_3$-$NO_x$ curve demonstrates superior reliability and is recommended for future studies.

## Supplement

Comparison of fitting significance amongst parametric models (Text S1); locations of the studied pollution monitoring stations (Figure S1); the $DPO_3$-$NO_2$ curves for BTH, FWP, YRD, PRD, Hong Kong, Macao and EU_US (2014-2019)

individually fitted by the studied models (Figures S2-S6 and S8-S9); the $DPO_3$-$NO_x$ curves for Hong Kong and EU_US (2014-2019) individually fitted by the studied models (Figures S7 and S10); the *p*-values per parameters in the convergent and effective $DPO_3$-$NO_x$ (or $NO_2$) fits based on the studied models (Figures S11-S12); variations of the $DPO_3$-$NO_2$ curves from 2014 to 2019 on the regional scale based on the CAQRA gridded data (Figure S13); spatiotemporal variations of the all-year MDA8-daytime-hour- and ozone-season midday specific ozone formation regimes (OFRs) from 2014 to 2019 in four city

agglomerations of China based on the *log-Bragg 3* model fitting $DPO_3$-$NO_2$ curves (Figures S14-S15); the fitting parameters, average levels (ppb) of $NO_2$ and $DPO_3$, partition points (ppb), transition points (ppb), as well as the proportions (%) respectively under the VOC-limited, transition and NOx-limited regimes specific to Figure 3 and Figure S13 (Tables S1 and S2).

## Author contributions

MH conceived and designed the research, collected and processed the data, developed the R code for the $DPO_3$-$NO_x$ (or $NO_2$) curve regression and $O_3$-$NO_x$-VOC sensitivity diagnosis, and drafted the manuscript; TL collected and processed the data, performed statistical analyses, and visualized the results.



**Code availability**

The data processes and visualization were conducted with R and Python, and the relevant code can be obtained upon request
to the corresponding author.

**Data availability**

The original data sources are detailed in Section 2.2, and the input data for figure visualization can be provided upon request
to the corresponding author.

**Competing interests**

The authors declare no competing financial interest.

**Acknowledgments**

The authors sincerely acknowledge Dr. Haichao Wang from School of Atmospheric Sciences at Sun Yat-sen University,
PR China, for his insights regarding the vertical variations of pollutants, which inspired us to give the possible reasons for the
discrepancy between our diagnostic results and the previous satellite-derived FNR-based results.

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
