# Peer review of "Technical note: Adaptably diagnosing O3-NOx-VOC sensitivity evolution with routine pollution and meteorological data"

_EGUsphere, 2025_

## Author Comment (AC1)

We appreciate the reviewer for the valuable comments and suggestions which have helped improve our manuscript. We addressed all of the specific comments individually. For clarity, the changes are **highlighted (in blue font) in the revised manuscript**. The itemized response/actions made to the manuscript are listed as below.

**Reviewer 1**

In "Technical Note: Adaptably diagnosing O3-NOx-VOC sensitivity evolution with routine pollution and meteorological data" Huang and Liao investigated the sensitivity of O3 formation at selected sites in China, the US and Europe by applying different fit equations to the datasets. The authors identify most of the studied regions to be dominated by VOC-limited O3 formation sensitivity.

While this is generally an important topic to investigate, unfortunately this study seems incoherent and is often difficult to follow. It remains largely unclear why and how the suggested fit equations are applied to the data and even more important what the added value of this analysis is. The in-situ observations investigated in this study can be used to directly infer the dominating sensitivity instead of using fit functions. Are any generalized conclusions drawn from the fitting? Could it be applied to other regions where observations are not available and how would that be possible considering that crossover points occur at NOx to VOC ratios that are characteristic to each location?

1. **Why** the suggested fit equations are applied to the data?

**As introduced in Section 1: Introduction,** the ozone level has increased in most urban areas worldwide, and the $O_3$-$NO_x$-VOC sensitivity has likely evolved in response to the divergent trends in precursor emissions. **Elucidating its long-term evolution is critical for effective ozone mitigation**.

However, those commonly used methods for ozone formation regime (OFR) diagnosis, such as the Empirical Kinetic Modelling Approach (EKMA) isopleth plot and chemical indicators (e.g., $H_2O_2/HNO_3$, $H_2O_2/NO_z$, etc.), heavily rely on observation-based or numerical models, constrained by limited field data and computational demands. They are typically applied in case studies (Sillman and He, 2002; Sillman and West, 2009; Xue et al., 2014; Ou et al., 2016; Li et al., 2018). Although the satellite-derived $HCHO/NO_2$ ratio (FNR)-based method enables the regional scale long-term $O_3$-$NO_x$-VOC sensitivity diagnosis (Jin and Holloway, 2015; Ren et al., 2022; Wang et al., 2021; Zhang et al., 2024), its fixed daily sampling time restricts insights into other hours, and the sensitivity always varies over time. **These constraints highlight the necessity for more flexible and adaptable approaches**.

**For a specific VOC reactivity (VOCR), the daytime ozone production (DPO$_3$) exhibits a characteristic skewed curve when plotted against NO$_x$ or NO$_2$ (Graphical Abstract),** which is transformed from the EKMA plot (Pusede and Cohen, 2012; Romer et al., 2018; Nussbaumer and Cohen, 2020; Guo et al., 2023; Yang et al., 2021). **On a DPO$_3$-NO$_x$ (or NO$_2$) curve (Graphical Abstract), the partition point and transition point are two key NO$_x$ (or**

**NO₂) levels for differentiating the O₃-NOₓ-VOC sensitivity.** The **partition point** is defined as the peak $DPO_3$ corresponding $NO_x$ (or $NO_2$) level, where the ozone formation equals to consumption, distinguishing the $NO_x$-limited/transition regime (to the left) and the VOC-limited regime (to the right); the **transition point** is defined as the $NO_x$ (or $NO_2$) level at the position indicating the onset of diminishing ozone production with respect to $NO_x$, which further differentiates the $NO_x$-limited and transition regimes. As referred to the study by Yang et al. (2021), we determined the transition point as the position with a half of the maximum curve slope.

**Although the in-situ observations investigated in this study can be used to directly infer the dominating sensitivity via non-parametric approach** (Huang et al., 2025)**, two limitations still persist about that method: (1)** a fixed smoothing span, the key configuration for non-parametric smoothing, failed to exhibit robustness in fitting performance across studied sites, which leads to uncertainty in determining the partition point and inhibits the adaptability of this method across a broader spatiotemporal range; **(2)** the non-parametric approach provided no information on the curve's height and width, which determine the transition point and vary with locations, study periods and environmental factors (e.g., temperature, VOCs, etc.). **The basic contour of the regular $DPO_3$-$NO_x$ (or $NO_2$) curve would not vary with the relative humidity, temperature, season, altitude, mixing layer height and VOC species** (Guo et al., 2023)**. This environmental stability makes it possible to be parametrically characterized. Therefore, seeking an effective empirical parametric model is necessary for more adaptably characterizing the $DPO_3$−$NO_x$ (or $NO_2$) relation and figuring out both the partition and transition points.** This is the most important objective of the present study.

2. **How** the suggested fit equations are applied to the data?

**As introduced in Section 2: Methodology,** the $DPO_3$-$NO_x$ (or $NO_2$) relation was regressed with the five-percentile-binned $NO_x$ (or $NO_2$) concentrations (or logarithms) and their corresponding average $DPO_3$ levels. The $DPO_3$ was defined as the difference between the MDA8-daytime (7:00-19:00 Local Time (LT)) hourly ozone concentration and the ozone concentration at 6:00 LT. A total of seven parametric models (Equations 1-7) were individually applied to characterize the $DPO_3$-$NO_x$ (or $NO_2$) relation. As in the bellowing **Response/Action 1-9** to Specific comment 1-9, we gave a more detail introduction of the rationale for selecting these studied models, which was added in the updated Supplement of the revised manuscript (Text S1).

3. What is the **added value** of this analysis?

In order to address the limitations of those commonly used diagnostic methods (**Line 39-46 in revised manuscript**) and the prior non-parametric fitting of $DPO_3$-$NO_x$ (or $NO_2$) curve (**Line 62-77 in the revised manuscript**), the present study aims to seek an effective empirical parametric model for more adaptably characterizing the $DPO_3$−$NO_x$ (or $NO_2$) relation and determining the dominating sensitivity of ozone formation. After a series of analyses as in Section 3.2: Which is the most capable parametric model, we identified that the *log-Bragg3*

model (Equation 3) performed the best.

Therefore, **one of the added values** of these analyses is that they make it **easier for OFR**
**diagnosis that can be adaptable to different locations and different time**, even though the
crossover points do not always occur at the same $NO_2$ mixing ratio. **This is particularly**
**important for elucidating the evolution of $O_3$-$NO_x$-VOC sensitivity on the large**
**spatiotemporal scale.** Furthermore, as discussed in Section 3.5: Implications of the *log-Bragg*
*3* model's parameters (*b, d*), **the identified model (*log-Bragg3* model, Equation 3) is also**
**able** to **provide implications of ozone formation intensity and the associated chemical**
**processes, indicating by its parameters.**

4. Are any **generalized conclusions** drawn from the fitting?

Based on the above responses, the generalized conclusion from the fitting is that the *log-Bragg3*
model (Equation 3) performed the best, compared with other models, in adaptably
characterizing the $DPO_3$−$NO_x$ (or $NO_2$) relation and diagnosing the dominating sensitivity of
ozone formation, which is also able to provide the implications of ozone formation intensity
and the associated chemical processes.

5. Could it be applied to other regions where observations are not available?

It could still be applied in other regions **where other reliable reanalysis data are available**,
even though there is no observation.

6. How would that be possible considering that crossover points occur at NOx to VOC ratios
that are characteristic to each location?

Yes, the crossover points are characteristic to different locations, and they theoretically depend
on local condition (e.g., VOCs or other relevant pollutants/radicals, meteorological factors,
etc.). However, our identified model can solve this problem by adaptably fitting the data at
different locations. **This is based on a hypothesis that the daytime ozone production ($DPO_3$)**
**exhibits a characteristic skewed curve when plotted against $NO_x$ or $NO_2$ for a specific**
**VOC reactivity (VOCR) (Line 47-48 in the revised manuscript).** And indeed, **as in Section**
**3.1, the above hypothesized $DPO_3^-$–$NO_x$ (or $NO_2$) relation was empirically validated**
**worldwide**, even in regions with severe $PM_{2.5}$ contamination, where the ozone formation is
additionally influenced by the aerosol-inhibited photochemical regime, such as BTH, FWP and
YRD in China (Ivatt et al., 2022; Geng et al., 2021; Kong et al., 2021; Xiao et al., 2022).

From this perspective, **the parameters in the identified model (*log-Bragg3*, Equation 3)**
**reflect the local condition to some extent**. For example, the fitting parameter *e* varied with
regions, as listed in Table S1 and Table S2; higher value indicates higher partition point (or
crossover point as referred by the reviewer). Furthermore, as discussed in Section 3.5, the
parameter *d* represents the maximum $DPO_3$ level, exhibiting higher ozone production with
higher value; and higher value of parameter *b* characterizes a steeper curve, indicating a
condition that favors faster change in ozone production efficiency for a given increment of $NO_x$.

It is further concerning that the authors have published a paper earlier this month (Huang et al., 2025), which they are now referring to have "critical limitations" which "fail" in respect to two different aspects (Line 54f.). This makes me wonder why the authors have not previously fixed these issues, considering that this previous paper was published one month after the submission of this manuscript.

We have clarified the distinctions and connections between the current study and our previous work (Huang et al., 2025), as outlined in the bellowing **Response/Action 1-1 to** Specific comment 1-1**.**

Some statements are further not backed with the current literature. The authors often use terms that are not commonly known in literature and do not provide sufficient definitions or explanations. The same applies to abbreviations that are not defined when first used. The figures have too many panels, are too small and have a low resolution, which makes them difficult to read and understand the results.

1. The issue regarding statements not backed with literatures was addressed as outlined in the bellowing **Response/Action 1-16** to Specific comment 1-16.

2. We speculate that the terms concerned by the reviewer, which are less frequently used in the literatures, might be the two terms "partition point" and "transition point". **The definitions as below were added in the revised manuscript (Line 55-62).**

"On a theoretically regular $DPO_3$-$NO_x$ (or $NO_2$) curve (Graphical Abstract), the partition point and transition point are two key $NO_x$ (or $NO_2$) levels for differentiating the $O_3$-$NO_x$-VOC sensitivity. The partition point is defined as the peak $DPO_3$ corresponding $NO_x$ (or $NO_2$) level, where the ozone formation equals to consumption, distinguishing the $NO_x$-limited/transition regime (to the left) and the VOC-limited regime (to the right); the transition point is defined as the $NO_x$ (or $NO_2$) level at the position indicating the onset of diminishing ozone production with respect to $NO_x$, which further differentiates the $NO_x$-limited and transition regimes. As referred to the study by Yang et al. (2021), we determined the transition point as the position with a half of the maximum curve slope in the present study."

3. All abbreviations were defined upon their first use in the revised manuscript, such as the term "OFR" as exemplified in bellowing **Response/Action 1-2** to Specific comment 1-2.

4. The Clearer figures were provided in the revised manuscript.

Considering these various drawbacks, unfortunately, I cannot recommend this manuscript for publication in its current state as it does neither meet the scientific nor the methodological standards of an ACP publication. If the authors wish to improve their manuscript in the future, please find more detailed comments and questions in the following, which might be helpful for revising the study.

**Specific comment 1-1:** Line 53 ff.: Could the authors describe the study of Huang et al., 2025?

What were the methods applied and the findings of this study?

**Response/Action 1-1:** Thank you for this valuable comment. The distinctions and connections
between the current study and our previous work (Huang et al., 2025), detailed as below, were
**added in the revised manuscript (Line 62-84).**

"In our previous study (Huang et al., 2025), the $DPO_3-NO_x$ (or $NO_2$) relation, as depicted in
the Scenario A, B or C within a completed skewed curve in Graphical Abstract, was proved
widespread based on the routine monitoring data in the Greater Bay Area, South China, which
was smoothed by a non-parametric regression technique. The smoothing curve was able to
effectively characterize the regional spatial pattern of $O_3$-$NO_x$-VOC sensitivity, differentiating
the ozone formation regimes (OFRs) in to the $NO_x$-limited/transition regime and the VOC-
limited regime, and was further utilized to examine temperature-dependent sensitivities. The
non-parametric approach is a commonly used method for smoothing a fluctuating numerical
series within local neighbourhoods, enabling the identification of intrinsic $DPO_3-NO_x$ (or $NO_2$)
relation. However, two limitations still persist: (1) a fixed smoothing span, the key
configuration for non-parametric smoothing, failed to exhibit robustness in fitting performance
across studied sites, which leads to uncertainty in determining the partition point and inhibits
the adaptability of this method across a broader spatiotemporal range; (2) the non-parametric
approach provided no information on the curve's height and width, which determine the
transition point and vary with locations, study periods and environmental factors (e.g.,
temperature, VOCs, etc.). As studied by Guo et al. (2023), the basic contour of the regular
$DPO_3$-$NO_x$ (or $NO_2$) curve would not vary with the relative humidity, temperature, season,
altitude, mixing layer height and VOC species. This environmental stability makes it possible
to be parametrically characterized. Therefore, seeking an effective empirical parametric model
is necessary for more adaptably characterizing the $DPO_3-NO_x$ (or $NO_2$) relation and figuring
out both the partition and transition points. This is the most important objective of the present
study.

However, it remains uncertain whether or not the regular $DPO_3-NO_x$ (or $NO_2$) relation is
globally prevalent. Therefore, it is essential to firstly verify the universality of this relation
using data from routine monitoring networks worldwide. Furthermore, based on the non-
parametric approach, our previous study (Huang et al., 2025) revealed that the applicability
and reliability for OFR diagnosis differed between the $DPO_3-NO_x$ and $DPO_3$-$NO_2$ curves at
several observation stations in Hong Kong. Accordingly, the present study also attempts to
compare the reliability between the two curves in diagnosing $O_3$-$NO_x$-VOC sensitivity across
a broader spatial range. "

**Specific comment 1-2:** Line 53: What is "OFR"? Please define abbreviations when first used.

**Response/Action 1-2:** Thank you for pointing it out. OFR is short for the term of ozone
formation regime. It has been defined **in the revised manuscript (Line 65-66).**

**Specific comment 1-3:** Line 54: The authors have published the study they are referring to
here (Huang et al., 2025) earlier this month and are now referring to critical limitations of their work. I find this a bit irritating. Why do the authors have not implemented the improvements
in the previous study?

**Response/Action 1-3:** The primary objective of our previous study (Huang et al., 2025) was
to investigate whether or not and how the OFRs shift with temperature in the Greater Bay Area,
South China. In this region, the theoretical $DPO_3-NO_x$ (or $NO_2$) curve was firstly convinced
regionally prevalent and able to effectively diagnose OFRs' spatial patten. This encourages us
to further verify whether this kind of curve is also globally widespread. In order to do so, we
have to spend more time to collect more pollution and meteorological monitoring data
worldwide and conduct the data pre-processing. Furthermore, the smoothing span, a key
configuration for non-parametric smoothing, failed to exhibit robustness in fitting performance
across studied sites, which leads to uncertainty in determining the partition point; therefore, we
had spent plenty of time to define the reasonable smoothing spans specific to the individual
studied observation stations and the relevant reanalysis data girds in that previous study. In a
similar way, we also have to spend sufficient time to firstly define the smoothing spans as
reasonable as possible for the stations in the present study, which is for further comparisons of
fitting performances between the empirical parametric models and the non-parametric
approach (as illustrated in Figures 1-3, Figures S2-S10, and Figures S13).

We do acknowledge that it would have been ideal by incorporating all potential improvements
into that previous study. However, the limitations of the previous work (outlined in Line 67-71
of the revised manuscript) did not hinder our ability to understand the temperature-related shift
of OFRs within a limited scope, such as the Greater Bay Area. However, in response to those
limits, the present study aims to find out a more adaptable method, the empirical parametric
modelling, for characterizing the theoretical $DPO_3-NO_x$ (or $NO_2$) curve and diagnosing OFRs.

**Specific comment 1-4:** Line 55 f.: How do the authors define the NOx-limited/transition
boundary? The transition point is commonly referred to as the crossover from NOx- to VOC-
sensitive chemistry, but there is no exact definition of a transition region in textbook literature.
If I read the graphical abstract correctly it is related to the 95[th] percentile of O3 production.
Where does this definition come from and what's the reasoning for it?

**Response/Action 1-4:** Thank you for pointing it out. The crossover from $NO_x$-
limited/transition regime to VOC-sensitive regime is the partition point, rather than the
transition point. The transition point is the crossover from NOx-limited regime to transition
regime, and it indicates the onset of diminishing ozone production with respect to $NO_x$. As
referred to the study by Yang et al. (2021), we determined the transition point as the $NO_x$ (or
$NO_2$) level corresponding to the position with a half of the maximum curve slope in the present
study. **The definition and determination of the two key points were added in the revised**
**manuscript Line (Line 54-59).**

**The relation between the 95[th] percentile of $O_3$ production and the transition point,**
**detailed as below, was added in the revised manuscript (Line 237-244).**

"According to the study by Yang et al. (2021), the transition point for the parametric fitting curve was defined as the $NO_x$ (or $NO_2$) concentration corresponding to the position with a half
of the maximum fitting curve slope (the blue dotted lines in Figures 3 and S13), after which
ozone formation became less dependent on $NO_x$ but significantly more dependent on VOCR.
This parametric transition point exactly corresponded to the $DPO_3$ level in the top 4.9% of the
*log-Bragg3* model predictions, so that the transition point for the non-parametric smoothing
curve was determined as the $NO_x$ (or $NO_2$) level corresponding to the top 4.9% smoothing
$DPO_3$ level (the red dotted lines in Figures 3 and S13)."

**Specific comment 1-5:** Line 56: What do the authors mean by parametric modeling? Is this a
reference to parameterizations in atmospheric models or something different?

**Response/Action 1-5:** The term "parametric modeling" refers to regression with empirical
parametric model (Equations 1-7), which is distinct from the parameterization schemes
incorporated in atmospheric models. By fitting data with empirical models, we can adaptively
obtain the parameters of these models that are specific to the studied locations. The theoretical
meanings of the models' parameters were described in Section 2.1. For instance, by fitting data
from different locations with the *log-Bragg3* model (Equation 3), we can obtain different sets
of fitting values for its three parameters ($b, d, e$). The fitting values of parameters $b$ and $d$
determine the transition point (Section 3.4) and respectively imply ozone production intensity
and the related chemical processes (Section 3.5), and the parameter $e$ corresponds to the
partition point (Section 3.4). Therefore, it is possible to conveniently compare the
characteristics of ozone formation amongst different locations using the *log-Bragg3* model
(Equation 3).

**Specific comment 1-6:** Line 57: What do the authors mean by environmental stability?
Whenever using non-textbook terms, I recommend a full definition and explanation.

**Response/Action 1-6:** The sentence "The $DPO_3$-$NO_x$ (or $NO_2$) curve shows environmental
stability (Guo et al., 2023), enabling the parametric characterization" was rewritten as "……the
basic contour of $DPO_3$-$NO_x$ (or $NO_2$) curve would not vary with the relative humidity,
temperature, season, altitude, mixing layer height and VOC species. This environmental
stability makes it possible to be parametrically characterized." **This is shown in Line 74-75**
**of the revised manuscript.**

**Specific comment 1-7:** Line 58: Please elaborate on the "bend" – what is it and where is it
coming from? The cited literature Romer et al., 2018 and Guo et al. 2023 do not seem to
mention / explain this bend. How can PO3 have two different values for the same NO2?

**Response/Action 1-7:** As referred to the Supplementary Information for Guo et al. (2023), the
$DPO_3$-$NO_2$ curve exhibited a bend at the end of the curve for some cases, especially for some
VOC species (like alkanes in Fig. S5) and when excluding the reaction of $NO+NO+O_2=2NO_2$
in box model (Fig. S6: C→(C)), while the $DPO_3$-$NO_x$ curve did not show such bending
behavior (Figure 2(b) in Romer et al. 2018).

The ends of both curves reflect the relatively low $DPO_3/NO_2$ ratio and high $NO_x$ level, and this typically indicates a condition that the reaction of OH with $NO_2$ dominates the fate of $HO_x$, slowing the oxidation of organic precursor, and gradually terminating the ozone production (Pusede et al., 2015; Romer et al., 2018). When applying the $DPO_3$-$NO_2$ curve, the ozone production might decrease with $NO_2$ under this condition, potentially leading to a pseudo diagnostic result indicative of a $NO_x$-limited regime under a realistic $NO_x$-saturated condition. In contrast, when applying the $DPO_3$-$NO_x$ curve, the ozone production continues to decline with the increasing $NO_x$ level under this low-$DPO_3$/$NO_2$-ratio condition, thereby diagnosed as the VOC-limited regime.

The above explanation was detailed in Section 3.3: Comparison of reliabilities between the $DPO_3$-$NO_2$ and $DPO_3$-$NO_x$ curves (**Line 216-227 in the revised manuscript).**

**Specific comment 1-8** Line 77 f.: What exactly are parametric vs non-parametric results?

**Response/Action 1-8:** The parametric results are the parametric fitting curves (the blue fitting curves in Figure 3, Figure S2-S10, and Figure S13) and their corresponding partition points (the blue dashed dotted vertical lines in Figure 3, Figure S2-S10, and Figure S13) , while non-parametric results referred to the non-parametric smoothing curves (the red smoothing curves in Figure 3, Figure S2-S10, and Figure S13) and their corresponding partition points (the red dashed dotted vertical lines in Figure 3, Figure S2-S10, and Figure S13).

For more clarity, the sentence "The parametric model validity was confirmed when its curve and partition point aligned well with the non-parametric results" was rewritten as "The parametric model validity was confirmed when its curve (the blue fitting curve in Figure 3, Figure S2-S10, and Figure S13) and partition point (blue dashed dotted vertical line in Figure 3, Figure S2-S10, and Figure S13) aligned well with those obtained from non-parametric approach (the red smoothing curve in Figure 3, Figure S2-S10, and Figure S13; red dashed dotted vertical line in Figure 3, Figure S2-S10, and Figure S13)" This is shown in **Line 99-102 of the revised manuscript**.

**Specific comment 1-9:** Line 80 ff.: Did the authors use these equations to fit their data? How were these fits chosen?

**Response/Action 1-9:** Yes, we use models (Equations 1-7) to fit our data, respectively. The rationale for selecting the studied models, detailed as below, were **added in the updated supplement (Text S1)**.

"The Equation 1 in the article text and the Equations S1-S4 provided here are usually used to describe a phenomenon where the Y variable increases to reach a maximum at a certain level of the X variable, and decreases afterwords. For example, they can be applied to determine the maximum growth rate of plant at its corresponding optimal temperature level, as well as in the cases related to bioassays in toxicology/biology study: low doses of exogenous substances induce irritation effects. Only Equation 1 in the article text is capable to describe a skewed and asymmetric curve with a maximum, whereas Equations S1–S4 provided here are limited to describing the normal and symmetric curves.

$$Y = d \times exp[-b \times (X - e)^2] \tag{S1}$$

$$Y = c + (d - c) \times exp[-b \times (X - e)^2] \tag{S2}$$

$$Y = \frac{d}{1+b \times (X-e)} \tag{S3}$$

$$Y = c + \frac{d-c}{1+b \times (X-e)} \tag{S4}$$

The *Poly2* model (Equation S5) provided as below can also be used to describe a symmetric curve and is ever applied in fitting the relation of $O_3$-HCHO/$NO_2$ ratio for diagnosing the $O_3$-$NO_x$-VOC sensitivity (Jin et al., 2020).

$$Y = b_0 + b_1 \times X + b_2 \times X^2 \tag{A5}$$

In the present study, the $DPO_3$-$NO_x$ (or $NO_2$) diagram is hypothesized to be a skewed and asymmetric curve with a maximum, and thus can be appropriately described by Equation 1. To explore more alternative fitting approaches, we attempted to reduce the skewness of the $DPO_3$-$NO_x$ (or $NO_2$) curve by logarithmizing the $NO_x$ (or $NO_2$) concentrations. Therefore, the Equations 2-7 in the article text are the transformed forms of the Equation 1 and Equations S1-S5 with the X-coordinate logarithmized."

**Specific comment 1-10:** Line 108: What is the study period?

**Response/Action 1-10:** Thank you for pointing it out. The study period was 2014-2019, which was **added in the revised manuscript (Line 136).**

**Specific comment 1-11:** Line 117: What are "records $\leq 4$"? Can the authors provide a reasoning for this "no precipitation" definition? I am not aware of being able to accurately infer rainfall from cloud cover.

**Response/Action 1-11:** Thank you for this question. The explanation, detailed as below, was **added in the revised manuscript (Line 144-151)**.

"The cloud records provided in the NOAA-Integrated Surface Database (ISD), obtained via the *R* package *worldmet*, range from 0 (representing no visible cloud cover) to 9 (representing a completely overcast sky). The cloud records with values $\leq 4$ indicate that $< 50\%$ of the sky is obscured by clouds. However, there are limited rainfall recordings in the ISD compared to cloud cover, especially in the US the studied regions in China (except Hong Kong). Based on the meteorological data for the Europe and Hong Kong, where both rainfall and cloud cover recordings are comprehensively available, the precipitation was significantly lower when $< 50\%$ of the sky is covered by clouds, compared to the instances where $> 50\%$ of the sky is obscured. Therefore, the "no precipitation" scenario for Europe and US was defined as the hours of 50% cloud cover with the records $\leq 4$ in ISD, rather than the zero-precipitation hours."

**Specific comment 1-12:** Line 130: The European Union does not describe a geographical region and some parts of the map in Figures S1 are not part of the EU. I recommend referring to the region as, e.g. Europe.

**Response/Action 1-12:** Thank you for this kind recommendation. **It was revised throughout the manuscript and the updated supplement.**

**Specific comment 1-13:** Figure 1: The panels are too small and the resolution too low. It is difficult to read the legend. It is further difficult to distinguish between any of the stations because the data points are overlapping.

**Response/Action 1-13:** Thank you for this comment. We have improved the resolution of this figure in the revised manuscript.

**Specific comment 1-14:** Line 170 ff.: It is unclear what exactly the authors are trying to show in Figure 1. What is a parametric y-axis and a non-parametric x-axis approach?

**Response/Action 1-14:** The y-axis represents the $NO_x$ (or $NO_2$) concentrations corresponding to the partition points obtained from the parametric models, while the x-axis represents those obtained from the non-parametric approach.

**Specific comment 1-15:** Line 174 f.: What are the definitions of the scenarios the authors are referring to?

**Response/Action 1-15:** As illustrated in the graphical abstract, taking the red curve specific to the lower VOC reactivity (VOCR1) as the example, the Scenario A is referred to as the curve portion within the yellow dashed box, while the Scenarios B and C corresponding to the curve portion within the green and blue dashed boxes, respectively. The non-parametric fitting can only feature one of the three scenarios based on the realistic data within a theoretically completed curve

**Specific comment 1-16:** Line 180 ff.: Is this the bend that the authors were referring to earlier? The reaction of OH + NO2 is a termination reaction of the HOx cycle and its dominance characterizes a VOC-limited O3 formation regime. Unlike the authors state, the cited studies do not show the existence of a pseudo NOx limited under a NOx saturated regime. Further evidence would be required to prove this statement of the authors, which does not agree with our current knowledge of O3 formation sensitivity.

**Response/Action 1-16:** Thank you for this comment. We do acknowledge that the reference citations presented here may lead to potential ambiguity in interpretation.

The statement "a pseudo diagnostic result indicative of a NOx-limited regime under a realistic NOx-saturated condition" is the finding derived from our present study, rather than the conclusions drawn from the cited references of Guo et al., 2023; Romer et al., 2018; Pusede et al., 2015. However, the studies by Guo et al. (2023) and Romer et al. (2018) provided the evidence regarding the diagnostic uncertainty associated with the application of $DPO_3$-$NO_2$
curve; while the studies by Pusede et al. (2015) and Romer et al. (2018) provide a possible
explanation for this kind of uncertainty.

More specifically, as referred to the **Supplementary Information for Guo et al. (2023),** the
$DPO_3$-$NO_2$ curve exhibited a bend at the end of the curve for some cases, especially for some
VOC species **(like alkanes in Fig. S5)** and when excluding the reaction of $NO+NO+O_2=2NO_2$
in box model **(Fig. S6: C→(C)),** while the $DPO_3$-$NO_x$ curve did not display such bending
behavior **(Figure 2(b) in Romer et al., 2018)**. The ends of both curves reflect the relatively
low $DPO_3$/$NO_2$ ratio and high $NO_x$ level, and this typically indicates a condition that the
reaction of OH with $NO_2$ dominates the fate of $HO_x$, slowing the oxidation of organic precursor,
and gradually terminating the ozone production (Pusede et al., 2015; Romer et al., 2018). When
applying the $DPO_3$-$NO_2$ curve, the ozone production might decrease with $NO_2$ under this
condition, potentially leading to a pseudo diagnostic result indicative of a $NO_x$-limited regime
under a realistic $NO_x$-saturated condition. In contrast, when applying the $DPO_3$-$NO_x$ curve, the
ozone production continues to decline with the increasing $NO_x$ level under the low-$DPO_3$/$NO_2$-
ratio condition, thereby diagnosed as the VOC-limited regime.

Based on the above interpretation, we have re-organized the discussion and citations **in the**
**revised manuscript (Line 214-227)** as below:

"The $DPO_3$/$NO_2$ ratios at these stations in Hong Kong ranged from 0.1 to 0.6, much lower than
other stations/grid (BTH: 1.1-4.0, FWP: 1.3-3.4, YRD: 1.4-4.5, PRD: 1.3-4.5, Macao: 2.1,
Europe region/US: 0.3-16.5, other stations in Hong Kong: 0.8-6.5). Such low $DPO_3$/$NO_2$ ratios,
accompanied by high $NO_x$ level, typically occur at the ends of both $DPO_3$-$NO_2$ and $DPO_3$-$NO_x$
curves. As referred to the Figures S5-S6 in Guo et al. (2023), the $DPO_3$-$NO_2$ curve was found
to exhibit a bend at the its end in certain cases, especially for specific VOC species (like alkanes)
and when the reaction of $NO+NO+O_2=2NO_2$ is excluded in box model, while the $DPO_3$-$NO_x$
curve did not display such bending behaviour (Romer et al., 2018). A low $DPO_3$/$NO_2$ ratio at
high $NO_x$ level typically indicates a condition that the reaction of OH with $NO_2$ dominates the
fate of $HO_x$, slowing the oxidation of organic precursor, and gradually terminating the ozone
production (Pusede et al., 2015; Romer et al., 2018). When applying the $DPO_3$-$NO_2$ curve, the
ozone production might decrease with $NO_2$ under this condition, potentially leading to a pseudo
diagnostic result indicative of a $NO_x$-limited regime under a realistic $NO_x$-saturated condition.
In contrast, when applying the $DPO_3$-$NO_x$ curve, the ozone production continues to decline
with the increasing $NO_x$ level under this low-$DPO_3$/$NO_2$-ratio condition, thereby diagnosed as
the VOC-limited regime. Hence, the $DPO_3$-$NO_x$ curve is considered more reliable for
diagnosing $O_3$-$NO_x$-VOC sensitivity at any $NO_x$ level, and it is recommended to check the
$DPO_3$/$NO_2$ ratio before employing the $DPO_3$-$NO_2$ curve. "

**Specific comment 1-17:** Line 193 ff. / Figure 3: What is the added value of these fits? It is
possible to determine the dominating sensitivity of O3 formation based on the observational
data of O3 and NO2. Why are the fits needed? The individual fit parameters are likely different
for each location, as the crossover does not always occur at the same NO2 mixing ratio (depending on the availability of VOCs).

**Response/Action 1-17:** In Figure 3, the blue solid curves represent the parametric fittings
based on the *log-Bragg3* model (Equation 3). This model was proved the best to adaptably
characterize both the $DPO_3$-$NO_2$ and $DPO_3$-$NO_x$ curves and determine the dominating
sensitivity of ozone formation based on the routine recordings of $O_3$ and $NO_2$ or $O_3$ and $NO_x$.

**One of the added values** of these fits shown in Figure 3 is that **they make it easier for OFR**
**diagnosis that can be adaptable to different locations and different time,** even though the
crossover points does not always occur at the same $NO_2$ mixing ratio. **This is particularly**
**important for elucidating the evolution of $O_3$-$NO_x$-VOC sensitivity on the large**
**spatiotemporal scale.**

Furthermore, as discussed in Section 3.5: Implications of the *log-Bragg 3* model's parameters
(*b, d*), **the other value** is that the parametric fits in Figure 3 **also provide some implications**
**of ozone formation intensity and the associated chemical processes,** indicated by the
parameters *b* and *d* (Table S1).

However, according to the comparison of diagnostic reliability between the $DPO_3$-$NO_2$ and
$DPO_3$-$NO_x$ curves (in Section 3.3), the use of $DPO_3$-$NO_2$ curve may introduce significant
uncertainty when the $DPO_3/NO_2$ ratio is excessively low, and it is recommended to evaluate
the $DPO_3/NO_2$ ratio prior to applying this curve. Therefore, before applying the $DPO_3$-$NO_2$
curve for OFR diagnosis on the regional scale (in Section 3.4), we firstly checked the
$DPO_3/NO_2$ ratios, ranging from 0.88 to 4.98 for our studied regions, which were at the median
levels compared with those stations of pseudo-diagnosis in Hong Kong (0.1-0.6). Even for the
European region with the lowest ratios amongst out studied regions, the $DPO_3$-$NO_2$ curve is
still applicable, where the diagnostic results agreed well between the $DPO_3$-$NO_2$ and $DPO_3$-
$NO_x$ curves. **In a word, it is conditional to determine the dominating sensitivity of ozone**
**formation based on the observational data of $O_3$ and $NO_2$.**

**Specific comment 1-18:** Line 217 ff.: How was the log-Bragg 3 fit chosen? Does it provide
the best result? How was this evaluated?

**Response/Action 1-18:** Yes, the *log-Bragg3* model (Equation 3) provided the best fitting result,
compared to other models (Equations 1-2 and 4-7). In the present study, the parametric model
validity was confirmed when its curve and partition point aligned well with those obtained
from non-parametric approach, which revealed the intrinsic $DPO_3$-$NO_x$ (or $NO_2$) relation by
smoothing a numerical series within local neighborhoods. The studied parametric models were
individually applied to regress the $DPO_3$-$NO_x$ (or $NO_2$) relation for all the studied stations and
the Macao grid (494 fits).

Firstly, we identified that the models of ***log-Bragg3, log-Bragg4, log-Lorentz3, log-Lorentz4***
***and log-Poly2*** (Equations 3-7) exhibited **the highest fitting convergence**, with all 494
parametric fits successfully converging. However, not all the convergent fits were able to
characterize the regular diagram as in Graphical Abstract to effectively partition the $O_3$-$NO_x$-

VOC sensitivity.

Secondly, we further observed that the models of *log-poly2* **(Equation 7),** *log-Bragg3*
**(Equation 3) and** *log-Lorentz3* **(Equation 5)** were able to regress **the largest number of**
**convergent and effective fits** (*log-poly2*: 142/142 $DPO_3$-$NO_x$ fits, 494/494 $DPO_3$-$NO_2$ fits;
*log-Bragg3*: 141/142 $DPO_3$-$NO_x$ fits, 490/494 $DPO_3$-$NO_2$ fits; *log-Lorentz3*: 141/142 $DPO_3$-
$NO_x$ fits, 489/494 $DPO_3$-$NO_2$ fits).

Although all the *log-Poly2* fits (Equation 7) were convergent and effective, quite certain portion
of them did not achieve the statistical significance ($p > 0.1$) (Figures S11-S12 (g)). Amongst
all models, **the** *log-Bragg3* **(Equation 3) and** *log-Lorentz3* **(Equation 5)** models **performed**
**the best fitting significance,** with over 95% of fits achieving the statistical significance ($p <$
0.1) (Figures S11-S12 (c, e)).

Furthermore, we compared the partition points identified between the parametric and non-
parametric fits. It also showed that only **the** *log-Bragg3* **(Equation 3) and** *log-Lorentz3*
**(Equation 5) models were able to identify the partition points for all fits under Scenario**
**B** as illustrated in Graphical Abstract (Figure 1 (c, e, j, l)).

Despite comparable performance in terms of amounts of convergent and effective fits, fitting
statistical significance, and ability to identify partition point between the *log-Bragg3* and *log-*
*Lorentz3* models, **the** *log-Bragg3* **model is finally preferred due to the generally inferior**
**statistical properties exhibited by** *Lorentz* **models** (Ratkowsky, 1990)

Technical:

**Specific comment 1-19:** Line 29.: Please check the author of this reference ("Collaborators").

**Response/Action 1-19:** Thank you for pointing it out. This citing was corrected as "GBD 2019
Risk Factors Collaborators, 2020" **(Line 30),** and the relevant reference was corrected as **"**GBD
2019 Risk Factors Collaborators.: Global burden of 87 risk factors in 204 countries and
territories, 1990-2019: a systematic analysis for the Global Burden of Disease Study 2019,
Lancet, 396, 1223-1249, 10.1016/s0140-6736(20)30752-2, 2020**." (Line 365) in the revised**
**manuscript.**

**Specific comment 1-20:** Line 49 f.: There seems to be a part of the sentence missing "As NOx
increases."

**Response/Action 1-20:** Thank you for pointing it out. This is a repetitive statement, and it was
removed in the revised manuscript.

**References:**

[revised manuscript text omitted]

---

## Author Comment (AC2)

We appreciate the reviewer for the valuable comments and suggestions which have helped improve our manuscript. We addressed all of the specific comments individually. For clarity, the changes are **highlighted (in blue font) in the revised manuscript.** The itemized response/actions made to the manuscript are listed as below.

**Reviewer 2**

Huang et al. established statistical fitting equations for data sets from China, the United States, and Europe, and studied the sensitivity of O3 formation to precursors. This is an important topic at present. The author claimed that this parametric methodology overcomes the complexities and resource constraints inherent in conventional methods, and is expected as a unified tool to facilitate global ozone mitigation under evolving precursor emission patterns and climate change. Unfortunately, I do not agree with this view. The main reasons are as follows:

The authors established a unified statistical fitting equation to fit the nonlinear relationship between ozone and precursors. After comparing multiple nonparametric regression models, the authors found that the log-Bragg3 model is universal. However, the authors did not explain why the log-Bragg3 model equation is universal. The model lacks the basic connotation of atmospheric chemistry and atmospheric physics, but is only a statistical equation obtained by fitting. In terms of mathematics, one can completely fit a very complex nonlinear relationship with high precision by constructing high-order statistical equations. This is just a process of approximating fitting by the infinite series method in mathematics. However, this operation has not made any essential improvement to the study of ozone sensitivity. It is just a mathematical technique. Even if, as the author says, common results have been obtained in the existing data of different cities, there is no guarantee that universal results will be obtained in the future.

**Response/Action 2-1: All of the seven studied empirical parametric models (Equations 1-7) are selected on the basis of the basic connotation of ozone formation chemistry**, rather than just a process of approximating fitting by the infinite series method in mathematics.

**Based on the Empirical Kinetic Modelling Approach (EKMA) isopleth plot, one of the most validated diagnostic methods, the daytime ozone production (DPO$_3$) exhibits a characteristic skewed curve when plotted against NO$_x$ or NO$_2$ for a specific VOC reactivity (VOCR) (Graphical Abstract). This type of skewed and asymmetric diagram describing the DPO$_3$−NO$_x$ (or NO$_2$) relation had been reported in previous atmospheric chemistry relevant studies** (Pusede and Cohen, 2012; Romer et al., 2018; Nussbaumer and Cohen, 2020; Guo et al., 2023; Yang et al., 2021). As shown in the Graphical Abstract, ozone production rises with NO$_x$ is insensitive to VOCR. Reducing NO$_x$ is more effective than controlling VOCs to mitigate ozone. As NO$_x$ rises, it reaches its maximum and became limited by both NO$_x$ and VOCR. This indicates that controlling either or both precursors can effectively reduce ozone acceleration. With further NO$_x$ increase, it grows with VOCR but declines with NO$_x$. Here, VOC control becomes the key for ozone mitigation, while NO$_x$ reduction potentially leads to increase in ozone pollution. On a DPO$_3$-NO$_x$ (or NO$_2$) curve, the partition point and transition point are two key $NO_x$ (or $NO_2$) levels for differentiating the $O_3$-$NO_x$-VOC
sensitivity. The partition point is defined as the peak $DPO_3$ corresponding $NO_x$ (or $NO_2$) level,
where the ozone formation equals to consumption, distinguishing the $NO_x$-limited/transition
regime (to the left) and the VOC-limited regime (to the right); the transition point is defined as
the $NO_x$ (or $NO_2$) level at the position indicating the onset of diminishing ozone production
with respect to $NO_x$, which further differentiates the $NO_x$-limited and transition regimes. As
referred to the study by Yang et al. (2021), we determined the transition point as the position
with a half of the maximum curve slope in the present study.

**The above basic connotation of ozone formation chemistry for the hypothesized skewed**
**and asymmetric $DPO_3$-$NO_x$ (or $NO_2$) curve was detailed in Section 1 of the revised**
**manuscript (Line 47-61).**

Among all studied models, the *log-Bragg3* model was identified as the best model for globally
characterizing the $DPO_3$-$NO_x$ (or $NO_2$) relation and diagnosing $O_3$-$NO_x$-VOC sensitivity (or
referred to as ozone formation regimes, OFRs), based on the routine recordings of $O_3$ and $NO_2$
or $O_3$ and $NO_x$. The **improvements** of this model to the study of ozone sensitivity include:

(1) This model effectively **addresses the limitations of those commonly used methods**
**(Line 39-46) as well as the prior non-parametric fitting of $DPO_3$-$NO_x$ (or $NO_2$)**
**curve (Line 62-77)**, both of which are inadequate for elucidating the evolution of $O_3$-
$NO_x$-VOC sensitivity on the large spatiotemporal scale; therefore, this model **makes it**
**easier for OFR diagnosis that can be adaptable to different locations and different**
**time**, even though the partition and transition points, two key diagnostic $NO_x$ (or $NO_2$)
levels, are not always the same at different locations or time.
(2) As discussed in Section 3.5, this model is also able to **provide implications of ozone**
**formation intensity and the associated chemical processes, indicating by its**
**parameters.**

The **universal results** of the present study include: **(1)** the **basic contour of $DPO_3$-$NO_x$ (or**
**$NO_2$) diagram is a skewed and asymmetric curve**, and was verified universally prevalent,
including regions with severe $PM_{2.5}$ contamination where ozone formation is additionally
influenced by aerosol-inhibited photochemical regime (**Line 171-174 in the revised**
**manuscript**); **(2) the *log-Bragg3* model (Equation 3)** performed the best in characterizing
this relation, adaptably resolves the $O_3$-$NO_x$-VOC sensitivity continuum, and provide
parametric insights into ozone formation intensity (*d*), associated chemical processes (*b*), and
the $O_3$-$NO_x$-VOC sensitivity partition threshold (*e*). **In fact, we have confirmed that this**
**model can also fit the data from other regions and periods very well.**

**In the future, it is expected to be able to adaptably obtained the parameters of the $DPO_3$-**
**$NO_x$ (or $NO_2$) curves and diagnose the $O_3$-$NO_x$-VOC sensitivities at different locations**
**and different time.**

In fact, since 2020, there have been many breakthroughs in the research on the sensitivity of
O3 formation to precursors. The city lockdown measures during the COVID-19 period provided natural experiments for the emission reduction of precursors. Some new insights have
even overturned the past understanding of atmospheric chemistry's sensitivity to ozone, and
thereby improved the simulation effect of atmospheric ozone generation. This paper only
focuses on research data before 2019, and it seems to deliberately avoid the adverse effects of
emission reduction on simulation during the epidemic. Therefore, it is difficult to believe that
the results of this study have achieved universal success. Are the study findings applicable
during the COVID-19 pandemic in 2020?

**Response/Action 2-2:** Yes, in addition to the stations and periods incorporated in the present
study, we have confirmed that such a skewed and asymmetric diagram of $DPO_3$-$NO_x$ (or $NO_2$)
relation is also prevalent in other regions (e.g., Taiwan) and periods (2020-2022), where the
data can also be fitted well by the *log-Bragg3* model.

The reason using the data before 2019 is that we aim to verify whether such a skewed and
asymmetric diagram of $DPO_3$-$NO_x$ (or $NO_2$) relation can also be observed in those regions
where the ozone formation is additionally influenced by the aerosol-inhibited photochemical
regime, and the $PM_{2.5}$ pollution during our studied period was significantly more severe in the
regions of BTH, FWP and YRD in China compared to other regions (e.g., the European region
and US).

The paper mentioned their past research results, but did not explain in depth the improvements
of this method.

**Response/Action 2-3:** Thank you for this valuable comment. The distinctions and connections
between the current study and our previous work (Huang et al., 2025) were **added in the**
**revised manuscript (Line 62-83).**

The non-parametric approach in our previous study is a commonly used method for smoothing
a fluctuating numerical series within local neighbourhoods, enabling the identification of
intrinsic $DPO_3^-NO_x$ (or $NO_2$) relation. However, **two limitations still persist: (1)** a fixed
smoothing span, the key configuration for non-parametric smoothing, failed to exhibit
robustness in fitting performance across studied sites, which leads to uncertainty in determining
the partition point and inhibits the adaptability of this method across a broader spatiotemporal
range; **(2)** the non-parametric approach provided no information on the curve's height and
width, which determine the transition point and vary with locations, study periods and
environmental factors (e.g., temperature, VOCs, etc.). As studied by Guo et al. (2023), the basic
contour of the regular $DPO_3$-$NO_x$ (or $NO_2$) curve would not vary with the relative humidity,
temperature, season, altitude, mixing layer height and VOC species. This environmental
stability makes it possible to be parametrically characterized. **Therefore, seeking an effective**
**empirical parametric model is necessary for more adaptably characterizing the**
**$DPO_3-NO_x$ (or $NO_2$) relation and figuring out both the partition and transition points.**
**This is the most important objective of the present study.**

Even though the $DPO_3-NO_x$ (or $NO_2$) relation, as depicted in the Scenario A, B or C within a
completed skewed curve in Graphical Abstract, was firstly convinced regionally prevalent and able to effectively diagnose OFRs' spatial patten in that previous study, **it remains uncertain**
**whether or not this kind of regular DPO$_3$−NO$_x$ (or NO$_2$) relation is globally prevalent.**
**Therefore, it is essential to firstly verify the universality of this relation using data from**
**routine monitoring networks worldwide**. Furthermore, based on the non-parametric
approach, our previous study revealed that the applicability and reliability for OFR diagnosis
differed between the DPO$_3$−NO$_x$ and DPO$_3$-NO$_2$ curves at several observation stations in Hong
Kong. **Accordingly, the present study also attempts to compare the reliability between the**
**two curves in diagnosing O$_3$-NO$_x$-VOC sensitivity across a broader spatial range.**

Many details of the paper are obscure and difficult to understand. Especially in the explanation
of professional terms and details of model parameters. At the same time, the visualization of
the graph is also very vague, which seriously affects normal reading.

**Response/Action 2-4:** Thank you for this valuable comment. We speculate that the key
professional terms concerned by the reviewer, might be the two words "partition point" and
"transition point". **The definitions as below were added in the revised manuscript (Line 55-**
**61).**

"On a theoretically regular DPO$_3$-NO$_x$ (or NO$_2$) curve (Graphical Abstract), the partition point
and transition point are two key NO$_x$ (or NO$_2$) levels for differentiating the O$_3$-NO$_x$-VOC sens
itivity. The partition point is defined as the peak DPO$_3$ corresponding NO$_x$ (or NO$_2$) level, wh
ere the ozone formation equals to consumption, distinguishing the NO$_x$-limited/transition regi
me (to the left) and the VOC-limited regime (to the right); the transition point is defined as th
e NO$_x$ (or NO$_2$) level at the position indicating the onset of diminishing ozone production wit
h respect to NO$_x$, which further differentiates the NO$_x$-limited and transition regimes. As refer
red to the study by Yang et al. (2021), we determined the transition point as the position with
a half of the maximum curve slope in the present study."

**The model parameters were detailed in Section 2.1. Additionally, the rationale for**
**selecting these studied models, detailed as below, was added in the updated Supplement**
**(Text S1)**.

"The Equation 1 in the article text and the Equations S1-S4 provided here are usually used to
describe a phenomenon where the Y variable increases to reach a maximum at a certain level
of the X variable, and decreases afterwords. For example, they can be applied to determine the
maximum growth rate of plant at its corresponding optimal temperature level, as well as in the
cases related to bioassays in toxicology/biology study: low doses of exogenous substances
induce irritation effects. Only Equation 1 in the article text is capable to describe a skewed and
asymmetric curve with a maximum, whereas Equations S1–S4 provided here are limited to
describing the normal and symmetric curves.

$$Y = d \times exp[-b \times (X - e)^2] \tag{S1}$$

$$Y = c + (d - c) \times exp[-b \times (X - e)^2] \tag{S2}$$

$$Y = \frac{d}{1 + b \times (X - e)} \tag{S3}$$

$$Y = c + \frac{d - c}{1 + b \times (X - e)} \tag{S4}$$

The *Poly2* model (Equation S5) provided as below can also be used to describe a symmetric
curve and is ever applied in fitting the relation of $O_3$-HCHO/$NO_2$ ratio for diagnosing the $O_3$-
$NO_x$-VOC sensitivity (Jin et al., 2020).

$$Y = b_0 + b_1 \times X + b_2 \times X^2 \tag{A5}$$

In the present study, the $DPO_3$-$NO_x$ (or $NO_2$) diagram is hypothesized to be a skewed and
asymmetric curve with a maximum, and thus can be appropriately described by Equation 1. To
explore more alternative fitting approaches, we attempted to reduce the skewness of the $DPO_3$-
$NO_x$ (or $NO_2$) curve by logarithmizing the $NO_x$ (or $NO_2$) concentrations. Therefore, the
Equations 2-7 in the article text are the transformed forms of the Equation 1 and Equations S1-
S5 with the X-coordinate logarithmized."

**Finally, the Clearer figures were provided in the revised manuscript**.

**References:**

Guo, J., Zhang, X., Gao, Y., Wang, Z., Zhang, M., Xue, W., Herrmann, H., Brasseur, G. P.,
Wang, T., and Wang, Z.: Evolution of Ozone Pollution in China: What Track Will It Follow?,
Environmental Science & Technology, 57, 109-117, 10.1021/acs.est.2c08205, 2023.

Huang, M., Feng, Z., and Liao, T.: Shift of surface O3-NOx-VOC sensitivity with temperature
in the Guangdong-Hong Kong-Macao Greater Bay Area, South China, Environmental
Pollution, 125974, https://doi.org/10.1016/j.envpol.2025.125974, 2025.

Nussbaumer, C. M. and Cohen, R. C.: The Role of Temperature and NOx in Ozone Trends in
the Los Angeles Basin, Environmental Science & Technology, 54, 15652-15659,
10.1021/acs.est.0c04910, 2020.

Pusede, S. E. and Cohen, R. C.: On the observed response of ozone to $NO_x$ and
VOC reactivity reductions in San Joaquin Valley California 1995–present, Atmos. Chem. Phys.,
12, 8323-8339, 10.5194/acp-12-8323-2012, 2012.

Romer, P. S., Duffey, K. C., Wooldridge, P. J., Edgerton, E., Baumann, K., Feiner, P. A., Miller,
D. O., Brune, W. H., Koss, A. R., de Gouw, J. A., Misztal, P. K., Goldstein, A. H., and Cohen,
R. C.: Effects of temperature-dependent NOx emissions on continental ozone production,
Atmos. Chem. Phys., 18, 2601-2614, 10.5194/acp-18-2601-2018, 2018.

Yang, L., Yuan, Z., Luo, H., Wang, Y., Xu, Y., Duan, Y., and Fu, Q.: Identification of long-term
evolution of ozone sensitivity to precursors based on two-dimensional mutual verification,
Science         of         The         Total         Environment,         760,         143401,
https://doi.org/10.1016/j.scitotenv.2020.143401, 2021.